# An Intelligent System-Based Coffee Plant Leaf Disease Recognition Using Deep Learning Techniques on Rwandan Arabica Dataset

Eric Hitimana [1,*], Omar Janvier Sinayobye [1], J. Chrisostome Ufitinema [2], Jane Mukamugema [2], Peter Rwibasira [2], Theoneste Murangira [3], Emmanuel Masabo [1], Lucy Cherono Chepkwony [4], Marie Cynthia Abijuru Kamikazi [1], Jeanne Aline Ukundiwabo Uwera [1], Simon Martin Mvuyekure [5], Gaurav Bajpai [6] and Jackson Ngabonziza [7]

1. Department of Computer and Software Engineering, University of Rwanda, Kigali P.O. Box 3900, Rwanda; sijaom2@gmail.com (O.J.S.); masabem@gmail.com (E.M.); abijurucyn@gmail.com (M.C.A.K.); alineuwera00@gmail.com (J.A.U.U.)
2. Department of Biology, University of Rwanda, Kigali P.O. Box 3900, Rwanda; jufitine@gmail.com (J.C.U.); jmugema@gmail.com (J.M.); rwibasira@gmail.com (P.R.)
3. Department of Computer Science, University of Rwanda, Kigali P.O. Box 2285, Rwanda; tmurangira@gmail.com
4. African Center of Excellence in Data Science, University of Rwanda, Kigali P.O. Box 4285, Rwanda; luciejs@gmail.com
5. Rwanda Agriculture Board, Kicukiro District, Rubilizi, Kigali P.O. Box 5016, Rwanda; msmartin202@gmail.com
6. Directorate of Grants and Partnership, Kampala International University, Ggaba Road, Kansanga, Kampala P.O. Box 20000, Uganda; gb.bajpai@gmail.com
7. Bank of Kigali Plc, Kigali P.O. Box 175, Rwanda; ngabojck@gmail.com
* Correspondence: e.hitimana@ur.ac.rw

**Abstract:** Rwandan coffee holds significant importance and immense value within the realm of agriculture, serving as a vital and valuable commodity. Additionally, coffee plays a pivotal role in generating foreign exchange for numerous developing nations. However, the coffee plant is vulnerable to pests and diseases weakening production. Farmers in cooperation with experts use manual methods to detect diseases resulting in human errors. With the rapid improvements in deep learning methods, it is possible to detect and recognize plan diseases to support crop yield improvement. Therefore, it is an essential task to develop an efficient method for intelligently detecting, identifying, and predicting coffee leaf diseases. This study aims to build the Rwandan coffee plant dataset, with the occurrence of coffee rust, miner, and red spider mites identified to be the most popular due to their geographical situations. From the collected coffee leaves dataset of 37,939 images, the preprocessing, along with modeling used five deep learning models such as InceptionV3, ResNet50, Xception, VGG16, and DenseNet. The training, validation, and testing ratio is 80%, 10%, and 10%, respectively, with a maximum of 10 epochs. The comparative analysis of the models' performances was investigated to select the best for future portable use. The experiment proved the DenseNet model to be the best with an accuracy of 99.57%. The efficiency of the suggested method is validated through an unbiased evaluation when compared to existing approaches with different metrics.

**Keywords:** coffee leaf diseases; arabica coffee; deep learning; VGG16; DenseNet

## 1. Introduction

In Rwanda, agriculture accounts for a third of the GDP (gross domestic product) and makes up most jobs (approximately 80%) [1]. Additionally, a significant source of export value, particularly from the production of tea and coffee, accounts for more than 20% of Rwanda's overall exports by value across all sectors: more than $100 million/year [2].



Coffee is a $60 million industry in Rwanda that is primarily supplied by small-holder growers in the country's several agroecological zones. Along with the supply chain, the estimated 350,000 farmers whose livelihoods depend on growing coffee face jeopardy [3]. Therefore, the government has a top priority for the future development of this cash crop for export. Among the varieties of coffee plants in Rwanda, coffee arabica is the one shown promising resistance to climate change.

Small-scale farmers are primarily responsible for cultivating coffee, utilizing farming methods that involve fragmented land and numerous small plots spread across hilly areas. Typically, farmers own around two to six plots, depending on the number of coffee trees in each plot. Due to the scattered nature of these plots and the distance between them and the farmers' homes, the frequency of plant and land management activities is reduced. In addition, the mix-up of different crops with coffee along with separate small farms contributes to the spread of coffee leaf diseases. To rearrange land usage patterns, the Ministry of Agriculture and Animal Resources is executing a policy for land consolidation. Apart from the land management policies, the local farmers working unprofessionally are recommended to work cooperatively. This exercise helps them to get support from government agencies, such as training, and other inputs impacting the high quality of coffee production [3].

It has been reported that one of the crops at risk from climate change and the spread of disease/pest infections is coffee [4]. Furthermore, these circumstances arise from a variety of fungal species and other causes. The disease-causing agents, present on the leaves or other parts of the tree, are highly transmissible and can rapidly spread if not promptly addressed. According to the study, approximately 10% of the global plant economy is currently being impacted by the destructive consequences of plant infections and infestations [5].

Coffee farmers in Rwanda, like those in other regions, face continuous threats from various pests and diseases [6]. While some of these problems are minor and have a limited impact on crop yield and quality, others, such as coffee berry disease, coffee leaf rust, and coffee wilt disease (tracheomycosis), pose significant dangers. These serious diseases can not only affect individual farmers, but also have a major economic impact on countries or regions heavily reliant on coffee for foreign exchange earnings [7]. For instance, coffee wilt disease has been present in Africa since the 1920s, but since the 1990s, there have been widespread and recurring outbreaks. This results in substantial losses in countries such as Uganda, where over 14 million coffee trees have been destroyed, as well as in the Democratic Republic of Congo [8,9]. Once this disease takes hold on a farm, it becomes extremely challenging to control. Since coffee is a perennial crop, certain pests and diseases can survive and multiply throughout the growing season, continuously affecting the coffee plants, although their populations and impact may vary over time [10]. Other pests and diseases may only attack coffee during periods when conditions are favorable. Regardless, the damage caused by these pests and diseases can be significant, affecting both crop yield and quality [11].

Some pests and diseases, such as the white coffee stem borer, coffee wilt disease, parasitic nematodes, and root mealy bugs, could kill coffee plants outright. On the other hand, pests, such as the coffee berry borer, green scales, leaf rust, and brown eye spot, may not directly kill the plants but can severely hinder their growth by causing defoliation, ultimately impacting the quality of the coffee berries [12].

The process of diagnosing plant diseases is complex and entails tasks such as analyzing symptoms, recognizing patterns, and conducting various tests on leaves. These procedures require significant time, resources, and skills to complete [13]. In many instances, an incorrect diagnosis can result in plants developing immunity or reduced susceptibility to treatment. The intricacy of plant disease diagnosis has led to a decrease in both the quantity and quality of crop yields among farmers [14].

The drawn-out process frequently results in a widespread infection with significant losses [15]. Coffee is one of the most well-known drinks in the world and might go extinct without conservation, monitoring, and seed preservation measures, according

to scientists. Global warming, deforestation, illness, and pests are all factors in the decline [16]. By implementing effective crop protection systems, early monitoring and accurate diagnosis of crop diseases can be achieved, which, in turn, can help prevent losses in production quality.

Recognizing various types of coffee plant diseases is of utmost significance and is deemed a critical concern. Timely detection of coffee plant diseases can lead to improved decision-making in agricultural production management. Infected coffee plants typically exhibit noticeable marks or spots on their stems, fruits, leaves, or flowers. Importantly, each infection and pest infestation leaf have distinct patterns that can be utilized for diagnosing abnormalities. The identification of plant diseases necessitates expertise and human resources. Moreover, the process of manually examining and identifying the type of plant infection is subjective and time-consuming. Additionally, there is a possibility that the disease identified by farmers or experts could be misleading at times [17]. As a result, the use of an inappropriate pesticide or treatment might occur during the evaluation of plant diseases, ultimately leading to a decline in crop quality and potentially causing environmental pollution.

The application of computer vision and artificial intelligence (AI) technologies has been expressed as instrumental tools in combating plant diseases [18–20]. There are multiple methods available to address the problem of detecting plant infections with the help of technologies, as the initial signs of infection manifest as various spots and patterns on leaves [21]. The introduction of machine learning and deep learning techniques has led to significant advancements in plant disease recognition, revolutionizing research in this field. These techniques have facilitated automatic classification and feature extraction, enabling the representation of original image characteristics. Moreover, the availability of datasets, GPU machines, and software supporting complex deep learning architectures with reduced complexity has made the transition from traditional methods to deep learning platforms feasible. CNNs have particularly gained widespread attention due to their remarkable capabilities in recognition and classification. CNNs excel in extracting intricate low-level features from images, making them a preferred choice for replacing traditional methods in automated plant disease recognition and yielding improved outcomes [22].

The research problem is based on the numerous efforts of government agencies and farmers in the use of manual methods to detect coffee diseases. In addition, a huge monetary effort is used to train farmers in coffee disease identification. However, the trained methods result in wrong findings [23]. To remedy the detected diseases, they may use the wrong pesticides, which do not treat the matter but affect environmental degradation.

This study aimed to develop and train five deep learning models on the collected dataset of coffee arabica leaves and determine the best model yielding the best results by leveraging pre-trained models and transferring knowledge approaches. The objective was to identify the most effective transfer learning technique for achieving accurate classification and optimal recognition accuracy in a multi-class coffee leaf disease context. The main contributions of this study are (1) to assess, collect, and classify the coffee leaves dataset in the Rwandan context; (2) to apply different data preprocessing techniques on the labeled data set; and (3) to determine the best transfer learning technique for achieving the most accurate classification and optimal recognition on multi-class plant diseases.

The remaining sections of the paper are structured as follows. Section 2 details the related works of this research. Section 3 outlines the materials and methods employed in this study. The findings and results are presented in Section 4. Section 5 delves into the discussion of the various experiments conducted. Finally, in Section 6, the research concludes by summarizing the key points and outlining potential future directions for research.

## 2. Related Works

Several methods have been suggested by researchers to achieve the precise detection and classification of plant infections. Some of these methods employ conventional image processing techniques that involve manual feature extraction and segmentation [24].

Among many methods, the use of K-means clustering for image leaf segmentation by extracting infected regions and later performing classification using a multi-class support vector machine is investigated [25]. The probabilistic neural network method was used to extract methodologies with statistical features on cucumber plant infection [26]. The preprocessing of images, from red, green, and blue (RGB) conversion to gray; HE; K-means clustering; and contour tracing is computed, and the results are used for classifications using support vector machine (SVM), K-NN, and convolutional neural networks (CNN). The experiment was carried out on tomato leaf infection detection [27] and grapes [28]. The automatic detection of leaf damage on coffee leaves has been conducted using image segmentation with Fuzzy C-means clustering applied to the V channel of the YUV color space image [29]. The automatic identification and classification of plant diseases and pests as well as the severity assessment, specifically focusing on coffee leaves in Brazil, is investigated. They targeted two specific issues: leaf rust caused by Hemileia vastatrix and leaf miner caused by Leucoptera coffee. Various image processing techniques were employed, including image segmentation using the K-means algorithm, the Otsu method, and the iterative threshold method, performed in the YCgCr color space. Texture and color attributes were calculated for feature extraction. For classification purposes, an artificial neural network trained with backpropagation and an extreme learning machine was utilized. The images utilized were captured using an ASUS Zenfone 2 smartphone (ZE551ML) with a resolution of 10 Megapixels (4096 × 2304 pixels). The database used in the study consisted of 690 images [30].

Moreover, the existing models heavily depend on manual feature engineering techniques, classification methods, and spot segmentation. However, with the advent of artificial intelligence in the field of computer vision, researchers have increasingly utilized machine learning [31] and deep learning [32] models to improve recognition accuracy significantly.

A CNN-based predictive model for classification and image processing in paddy plants is proposed [33]. Similarly, the utilization of a CNN for disease detection in paddy fields using convolutional neural networks with four to six layers to classify various plant species is elaborated on [34]. The application of CNN with a transfer learning approach to classify, recognize, and segment different plant diseases is tested [35]. Although CNNs have been extensively used with promising results, there is a lack of diversity in the datasets employed [36]. To achieve the best outcomes, training deep learning models with larger and more diverse datasets is crucial. While previous studies have demonstrated significant achievements, there is still room for improvement in terms of dataset diversity, particularly in capturing realistic images from actual agricultural fields with diverse backgrounds.

Deep-learning models based on CNNs have gained popularity in image-based research due to their effectiveness in learning intricate low-level features from images. However, training deep CNN layers can be computationally intensive and challenging. To address these issues, researchers have proposed transfer learning-based models [37–39]. These models leverage pre-trained networks, such as VGG-16, ResNet, DenseNet, and Inception [40], which have been well-established and widely used in the field. Transfer learning allows for the models to leverage the knowledge gained from pre-training on large datasets, enabling faster and more efficient training on specific image classification tasks.

The focus of the automatic and accurate estimation of disease severity to address concerns related to food security, disease management, and yield loss prediction was investigated on beans [41]. They applied deep learning techniques to analyze images of Apple black rot from the Plant Village dataset, which is caused by the fungus Botryosphaeria obtusa. The study compared the performance of different deep learning models, including VGG16, VGG19, Inception-v3, and ResNet50. The results demonstrated that the deep VGG16 model, trained with transfer learning, achieved the highest accuracy of 90.5% on the hold-out test set.

The classification of cotton leaves based on leaf hairiness (pubescence) used a four-part deep learning model named HairNet. HairNet demonstrated impressive performance,

achieving 89% accuracy per image and 95% accuracy per leaf. Furthermore, the model successfully classified the cotton based on leaf hairiness, achieving an accuracy range of 86–99% [42]. A deep learning approach was developed to automate the classification of diseases in banana leaves. The researchers utilized the LeNet architecture, a CNN through a 3700 image dataset. The implementation of the approach utilized deeplearning4j, an open-source deep-learning library that supports GPUs. The experiment was applied to detect two well-known banana diseases, namely Sigatoka and Speckle [35].

The application of emerging technologies, such as image processing, machine learning, and deep learning in the agriculture sector, is transforming the industry, leading to increased productivity, sustainability, and profitability while reducing environmental impact. A lot of authors have investigated different algorithms for different or specific plant types to ensure common solutions; however, the solution is problem-specific. It has been observed that most of the modeling has been attempted on the Plant Village dataset [43] to check the performance of the models selected.

Table 1 showcases different methods used for plant leaf classification, along with the corresponding accuracy percentages achieved on different types of leaves. The "Proposed model" refers to DenseNet, which obtained an accuracy of 99.57% on coffee leaf classification.

**Table 1.** Comparison of our resulting model with existing deep learning models.

| Ref. No and Year | Method | Accuracy (%) | Plant Name |
|---|---|---|---|
| [44]—2021 | Proposed FCNN & SCNN Hybrid Principal | 92.01 | Crop Leaf |
| [45]—2021 | Component Analysis | 95.10 | Plant Leaf |
| [46]—2020 | Hybris PCA & Optimization Algorithm | 90.20 | Olive Leaf |
| [47]—2020 | ResNet50 | 99.00 | Okra Leaf |
| [48]—2020 | Deep CNN | 98.00 | Coffee Leaf |
| [49]—2022 | Deep Transfer EffientNet | 98.70 | Grape Leaf |
| Proposed model | DenseNet | 99.57 | Coffee Leaf |

## 3. Materials and Methods

For proper plant disease management, early detection of diseases in coffee leaves is required to facilitate farmers. This section provides a complete description of the methodology used to collect coffee leaves and the methods used to experiment with the modeling techniques. Discussion of the process to collect leaves and several transfer-learning algorithms have been elaborated on to investigate the best model responding to the research scope. The architecture and training process of each model with the experimental setup on the used dataset is also discussed.

Rwanda has many high mountains and steep-sloped hills, with much of the farmland suffering from moderate to severe soil erosion, and the appearance of coffee diseases and pest are based on climate variability [50]. Among different types of coffee plants, such as arabica and robusta [51], this study focuses on the most popular variety known as arabica [52] that exists in Rwanda. We surveyed and visited 10 coffee washing stations located in different 5 districts, such as Ngoma, Rulindo, Gicumbi, Rutsiro, and Huye. The districts selected represent all 27 districts of Rwanda caring about climate variations [50]. In each district, we sampled 30 farmers giving 150 sample sizes. The visit was done to cooperate with agronomists who know the coffee pests and diseases to support coffee leaf labeling activities and to engage farmers to assess if they have the capacity to identify different coffee leaf diseases. The visit was attempted in the harvesting session, which is in March 2021, and in the summer session, which is in June and July 2021. The dataset images were collected from four distinct provinces located in the Eastern (sunny region with high low altitude with no hills), Northern (the cold region with high altitude), Southern (the cold region with modulated altitude), and Western (cold, highlands with high altitude). The quantitative and qualitative methodology was adopted to investigate the disease occurrence distribution in Rwanda as shown in Figure 1.

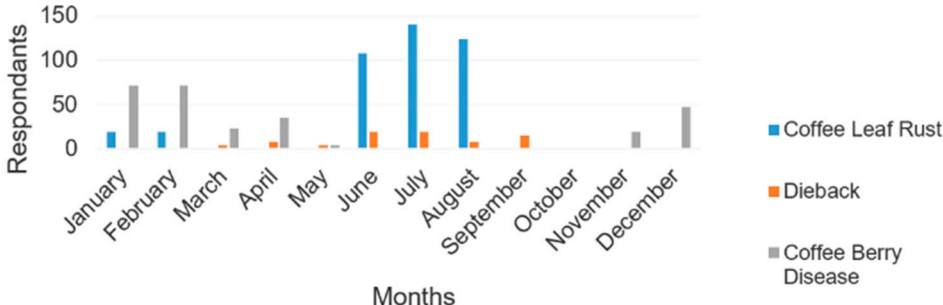

**Figure 1.** Knowledge of coffee diseases alongside disease occurrence in months.

According to our respondents, coffee leaf rust, known as "coffee leaf rust", is the most dangerous disease ravaging coffee in Rwanda. As shown in Figure 1, the disease occurs mostly in June, July, and August.

The process of data collection was followed by the experiment of coffee disease detection using deep learning techniques. Figure 2 details the architectural flow of the implementation.

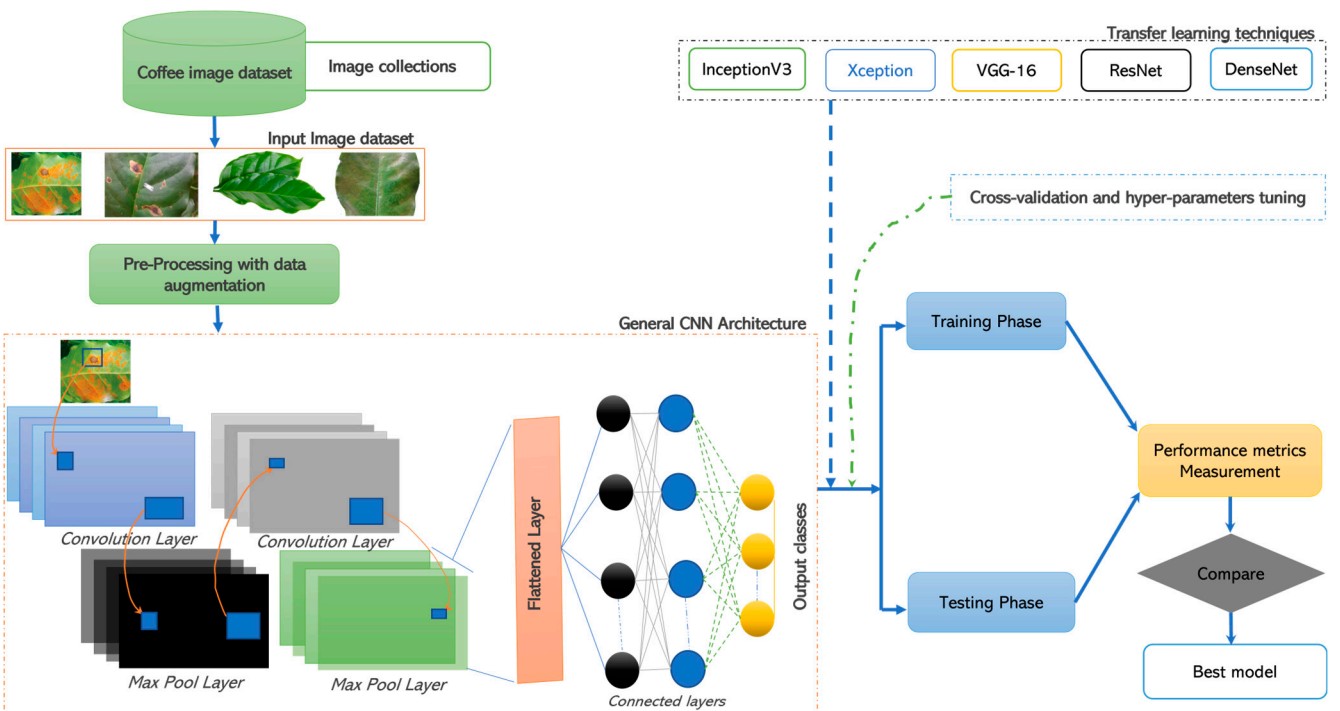

**Figure 2.** Proposed architectural implementation flow.

The suggested pipeline for detecting coffee leaf diseases begins with preparing the dataset and concludes with making predictions using different models and comparison analysis. To accomplish this, the Python 3.10 programming language, TensorFlow 2.9.1, numpy version 1.19.2, and matplotlib version 3.5.2 libraries were employed for dataset preparation and development environment setup. Those tools have proven to be useful for data preprocessing and modeling purposes [53,54]. The experiment used CNN deep learning models, such as InceptionV3, Resnet50, VGG16, Xception, and DenseNet models. The experiment used infrastructure with an HP Z240 workstation equipped with two Intel(R) Xeon(R) Gold 6226R and Tesla V100s 32GB memory NVIDIA GPU of 64 cores in total, which significantly accelerated the training process of deep neural networks. In the subsequent sections, each stage of the proposed coffee plant leaf disease detection pipeline will be thoroughly discussed.

*3.1. Dataset*

The researchers collected 37,939 images dataset in RGB format. The coffee images had at least four classes in the dataset, namely the class rust, red spider mite, miner, and healthy. The dataset's classes were made up of these directories, each of which corresponded to a certain disease.

Figure 3 shows the details of the sample dataset classes used in the experiment. Due to the severity of the matter, in a specific class, you may find different images with similar infections at different stages. This is because, at a certain stage, the model can be able to track and classify the real name or approximate name of the diseases.

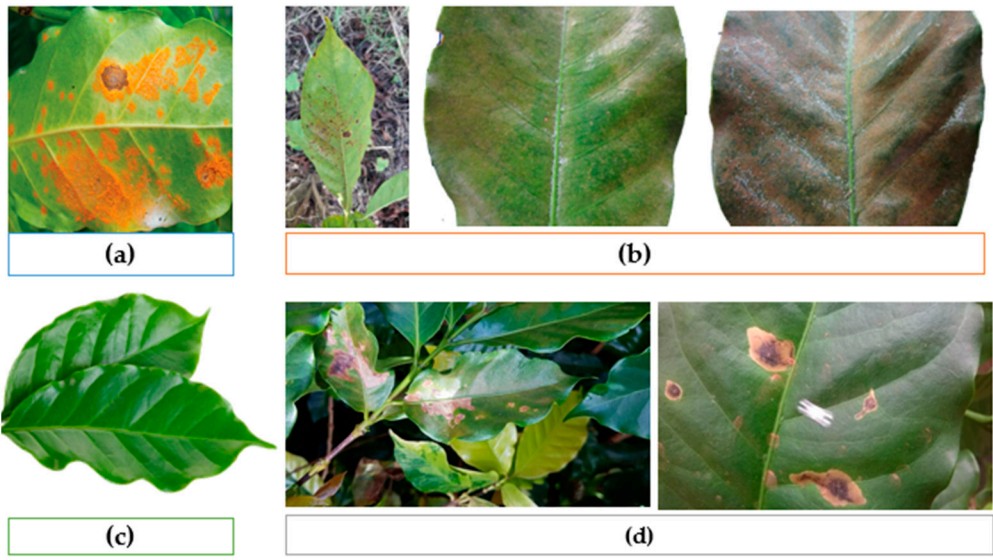

**Figure 3.** Few sampled coffee leaf image datasets. (**a**) Rust infection; (**b**) Red spider mites' infection in different stages; (**c**) Health leaf; (**d**) Miner infections.

Before supplying the images from the dataset to the CNN architectures, we preprocessed them to make sure the input parameters matched the requirements of the CNN model. Each input image was downsized to 224 × 224 dimensions after preprocessing. To guarantee that all the data were described under the same distribution, normalization (i.e., image/255.0) was then applied, which improved training convergence and stability [55].

*3.2. Used Deep Learning Models*

In the following section, this study details all the different models and tools used. The modeling of the coffee leaf images was conducted using different deep-learning techniques, such as InceptionV3, Resnet50, VGG16, Xception, and DenseNet as shown in Figure 2.

3.2.1. InceptionV3

InceptionV3, developed by Google Research, belongs to the Inception model series, and serves as a deep convolutional neural network structure. Its primary purpose is to facilitate image recognition and classification assignments [56–58].

Its architecture is known for its deep structure and the use of Inception modules. These modules consist of parallel convolutional layers with different filter sizes, allowing the network to capture features at multiple scales. By incorporating these parallel branches, the model can effectively handle both local and global features in an image [59]. One of the key innovations in InceptionV3 is the use of 1 × 1 convolutions, which serve as bottleneck layers. These 1 × 1 convolutions help reduce the number of input channels and computational complexity, making the network more efficient.

The Inception V3 model consists of a total of 42 layers, surpassing the layer count of its predecessors, Inception V1 and V2. Nonetheless, the efficiency of this model is

remarkable [60]. It can be fine-tuned on specific datasets or used as a feature extractor in transfer learning scenarios, where the pre-trained weights are utilized to extract meaningful features from images and train a smaller classifier on top of them. With its powerful deep learning architecture that excels in image recognition and classification tasks, this model was selected in this study to investigate its performance.

### 3.2.2. ResNet50

This research experiment suggested the use of ResNet-50 as Residual Network-50 introduced by Microsoft Research [61]. It is a variant of the ResNet family of models, which are renowned for their ability to train very deep neural networks by mitigating the vanishing gradient problem. It is known for its residual connection enabling the network to learn residual mappings instead of directly learning the desired underlying mapping. The residual connections facilitate passing information from earlier layers directly to later layers, helping to alleviate the degradation problem caused by increasing network depth.

The ResNet-50 architecture consists of 50 layers, including convolutional layers, pooling layers, fully connected layers, and shortcut connections. It follows a modular structure, where residual blocks with varying numbers of convolutional layers are stacked together [62]. Each residual block includes a set of convolutional layers, followed by batch normalization and activation functions, with the addition of the original input to the block. This ensures that the gradient flows through the skip connections and facilitates the learning of residual mappings.

The model was applied to plant disease detection [63,64] by extracting contextual dependencies within images, focusing on essential features of disease identification. The method was chosen to take advantage of its learning of residual mappings and feed the model with the coffee image classes and their features. The pre-training enables the model to learn generic visual features that can be transferred to different image-related tasks.

### 3.2.3. VGG16

The Visual Geometry Group 16 (VGG16) is a convolutional neural network architecture developed by the Visual Geometry Group at the University of Oxford. It is known for its simplicity and effectiveness in the image classification tasks model [65].

The VGG16 architecture consists of 16 layers, including 13 convolutional layers and 3 fully connected layers. It follows a sequential structure, where convolutional layers are stacked together with max pooling layers to progressively extract features from input images. The convolutional layers use small $3 \times 3$ filters, which help capture local patterns and details in the images [66]. The architecture maintains a consistent configuration throughout the network, with the number of filters increasing as the spatial dimensions decrease. This uniformity simplifies the implementation and enables the straightforward transfer of learned weights to different tasks [67].

The pre-training model of VGG16 enables the model to learn general visual representations, fine-tuned or used as feature extractors for specific tasks. Its deep structure and small receptive field have been considered in this research context to capture hierarchical features in coffee leaf images and avail all possible found classes.

### 3.2.4. Xception

Detailed as Extreme Inception, a deep convolutional neural network architecture introduced by François Chollet, the creator of Keras [68]. The model is based on the Inception architecture but incorporates key modifications to improve its performance and efficiency. Its architecture aims to enhance the depth-wise separable convolutions introduced in Inception modules. In depth-wise separable convolutions, the spatial convolution and channel-wise convolution are decoupled, reducing the number of parameters and computational complexity.

The architecture of Xception introduces the notation of an extreme version of Inception, where the traditional convolutional layer is replaced by a depth-wise separable convolution.

The extreme version of the Inception module enables it to capture spatial and channel-wise information more effectively. Xception has been pre-trained on large-scale image classification datasets, such as ImageNet, and has demonstrated impressive performance in various computer vision tasks [69].

It is used as a feature extractor or fine-tuned on specific datasets, enabling it to generalize well to various image-related tasks.

### 3.2.5. DenseNet

Dense Convolutional Network is a deep convolutional neural network architecture known for its dense connectivity pattern and efficient parameter sharing [70]. This sharing facilitates feature reuse and gradient flow throughout the network. It uses the concept of dense blocks, where each layer is connected to every other layer in a feed-forward manner. DenseNet takes this concept further by concatenating feature maps from all previous layers. This dense connectivity pattern enables direct connections between layers at different depths, facilitating the flow of information and gradients through the network [71].

The DenseNet architecture consists of dense blocks followed by transition layers. A dense block is a series of convolutional layers, where each layer's input is concatenated with the feature maps of all preceding layers. Transition layers are used to down-sample feature maps and reduce spatial dimensions. This architecture enables the model to capture both local and global features effectively.

The operational mechanism of a dense block as shown in Figure 4, supports the subsequent layers by applying batch normalization (BN), ReLu activation, convolution, and pooling to modify the outcome. It has achieved state-of-the-art results on various image classification benchmarks. In the coffee leaves context, the DenseNet model has been used to classify the leaf based on the list of trained dataset classes.

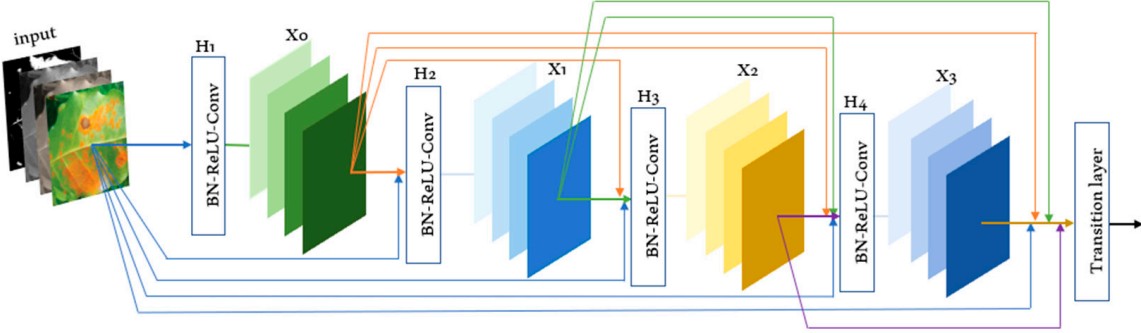

**Figure 4.** DenseNet Architecture.

### *3.3. Performance Measurement*

The experimental setup has been conducted using the methodology, methods, and infrastructures discussed in the above sections. To measure the performance of the transfer learning techniques, different metrics were considered. The performance accuracy matrix, precision-recall metric, and receiver operating characteristic (ROC), with the area under the curve (AUC), are being used to evaluate segmentation performance. The performance of the classifier is measured using evaluation metrics to select the best-performing ones for further use.

### 3.3.1. Precision-Recall Curve

The confusion matrix is a useful tool for assessing performance by comparing actual and predicted values. It provides insights into sensitivity, which represents the true positive rate and indicates the ability to correctly identify healthy and diseased leaves. Precision–recall curves are used in binary classification to study the output of a classifier.

To extend the precision–recall curve and average precision to multi-class or multi-label classification, it was necessary to binarize the output. One curve could be drawn per label, but one could also draw a precision–recall curve by considering each element of the label indicator matrix as a binary prediction (micro-averaging).

$$\text{Precision} = \frac{\text{True positives}}{\text{True Positives} + \text{False Positives}} \tag{1}$$

$$\text{Recall} = \frac{\text{True positives}}{\text{True Positives} + \text{False Negatives}} \tag{2}$$

The performance evaluation of plant disease classification involved analyzing the output, which could be binary or multiclass. Specificity, accuracy was referred to as the positive predicted value and defined in Equation (1). Recall, also known as the probability of detection, was calculated by dividing the number of correctly classified positive outcomes by the total number of positive outcomes (Equation (2)).

### 3.3.2. Receiver Operating Characteristic (ROC) Curve

The curve is mainly used to understand deterministic indicators of categorization sorting and computational modeling issues. ROC curves feature true positive rate (TPR) on the Y axis and false positive rate (FPR) on the X axis. The meaning is that the top left corner of the plot is the "ideal" point—a FPR of zero and a TPR of one. This is not very realistic, but it does mean that a larger area under the curve (AUC) is usually better. The "steepness" of ROC curves is important since it is ideal to maximize the TPR while minimizing the FPR. ROC curves are typically used in binary classification, where the TPR and FPR can be defined unambiguously.

Average precision (AP) summarizes such a plot as the weighted mean of precisions achieved at each threshold, with the increase in recall from the previous threshold used as the weight:

$$\text{AP} = \sum_{n=0}^{n} (R_n - R_{n-1})P_n \tag{3}$$

where $P_n$ and $R_n$ are the precision and recall at the nth threshold. A pair $(R_k, P_k)$ is referred to as an operating point. AP and the trapezoidal area under the operating points are calculated using the function sklearn.metrics.auc of Python package to summarize a precision–recall curve that led to different results.

### 3.3.3. Matthews Correlation Coefficient (MCC)

As an alternate approach that is not influenced by the problem of imbalanced datasets, the Matthews correlation coefficient is a technique involving a contingency matrix. This method calculates the Pearson product-moment correlation coefficient [72] between predicted and actual values. It is expressed in Equation (4) where TP is true positive.

$$\text{MCC} = \frac{\text{TP} \times \text{TN} - \text{FP} \times \text{FN}}{\sqrt{(\text{TP} + \text{FP}) \times (\text{TP} + \text{FN}) \times (\text{TN} + \text{FP}) \times (\text{TN} + \text{FN})}} \tag{4}$$

(Worst value: −1; best value: +1)

MCC stands out as the sole binary classification measure that yields a substantial score solely when the binary predictor effectively predicts most of the positive and negative data instances accurately [73]. It assumes values within the range of −1 to +1. The extreme values of −1 and +1 signify completely incorrect classification and flawless classification, respectively. Meanwhile, MCC = 0 is the anticipated outcome for a classifier akin to tossing a coin.

### 3.3.4. F1 Scores

Among the parametric group of F-measures, which is named after the parameter value β = 1, the F1 score holds the distinction of being the most frequently employed metric. It is determined as the harmonic average of precision and recall (refer to the formulas (1) and (2)), and its shape is expressed in the Equation (5):

$$\text{F1 score} = \frac{2 \times \text{TP}}{2 \times \text{TP} + \text{FP} + \text{FN}} \tag{5}$$

(Worst value: −1; best value: +1)

The F1 score spans the interval [0, 1], with the lowest value achieved when TP (true positives) equals 0, signifying the misclassification of all positive samples. Conversely, the highest value emerges when FN (false negatives) and FP (false positives) both equal 0, indicating flawless classification. There are two key distinctions that set apart the F1 score from MCC and accuracy: firstly, F1 remains unaffected by TN (true negatives), and secondly, it does not exhibit symmetry when classes are swapped.

## 4. Results

In this study, each experiment involved evaluating the training accuracy and testing accuracy. The losses incurred during the testing and training phases were computed for every model. The collected coffee leaves dataset was utilized to train the DCNN with transfer learning models. The selected pre-trained models are ResNet-50, Inception V3, VGG-16, Xception, and DenseNet.

### 4.1. Description of Dataset

To conduct our experimental analysis, the dataset was partitioned into three subsets: training samples, testing samples, and validation samples. Among the coffee plant leaf disease classes, a total of 37,939 images were available and trained with a ratio of 80:10:10. Out of these, 30,053 samples were used for training, 3793 for validation, and 4093 for testing. It is important to note that all these sets, including the training, testing, and validation sets, encompassed all four classes representing coffee plant leaf diseases used in this research context.

### 4.2. Preprocessing and Data Augmentation

The dataset consisted of four diseases of one type of crop species (coffee arabica). For our experimental purposes, we utilized color images from the collected dataset, as it was shown that they aligned well with the transfer learning models. To ensure compatibility with different pre-trained network models that require varying input sizes, the images were downscaled to a standardized format of 256 × 256 pixels. For VGG-16, DenseNet-121, Xception, and ResNet-50, the input size was set to 224 × 224 × 3 (height, width, and channel depth) while for Inception V3, the input shape was 299 × 299 × 3.

Although the dataset contained many images, approximately 37,939, depicting various coffee leaf diseases, these images accurately represented real-life images captured by farmers using different image acquisition techniques, such as high-definition cameras and smartphones, and downloaded from the internet. Due to the substantial size of the dataset, there was a risk of overfitting. To overcome the overfitting, regularization techniques were employed, including data augmentation after preprocessing.

In order to maintain the data augmentation capabilities, this study applied several transformations to the preprocessed images. Those transformations include clockwise and anticlockwise rotation, horizontal and vertical flipping, zoom intensity, and rescaling of the original images. This technique not only prevented overfitting and reduced model loss, but also enhanced the model's robustness, resulting in improved accuracy when tested with the real-life coffee plant images.

### 4.3. Network Architecture Model

The selection of pre-trained network models was based on their suitability for the task of plant disease classification. Detailed information about the architecture of each model can be found in Table 2. These models employ different filter sizes to extract specific features from the feature maps. The filters play a crucial role in the process of feature extraction. Each filter, when convolved with the input, extracts distinct features, and the specific features extracted from the feature maps depend on the values assigned to the filters. This research experiment utilized the original pre-trained network models, incorporating the specific combinations of convolution layers and filter sizes employed in each model.

**Table 2.** Pre-trained network architecture models' parameters.

| Parameters | InceptionV3 | Xception | ResNet50 | DenseNet | VGG16 |
|---|---|---|---|---|---|
| Total layers | 314 | 135 | 178 | 430 | 22 |
| Max pool layers | 4 | 4 | 1 | 1 | 5 |
| Dense layers | 2 | 2 | 2 | 2 | 2 |
| Drop-out layers | - | - | 2 | - | 2 |
| Flatten layers | - | - | 1 | - | 1 |
| Filter size | $1 \times 1, 3 \times 3, 5 \times 5$ | $3 \times 3$ | $3 \times 3$ | $3 \times 3, 1 \times 1$ | $3 \times 3$ |
| Stride | $2 \times 2$ | $2 \times 2$ | $2 \times 2$ | $2 \times 2$ | 1 |
| Trainable parameters | 23,905,060 | 22,963,756 | 25,689,988 | 8,091,204 | 15,244,100 |

Table 2 provides various parameters for different network models, including InceptionV3, Xception, ResNet50, VGG16, and DenseNet. The parameters include the total number of layers, max pool layers, dense layers, dropout layers, flatten layers, filter size, stride, and trainable parameters. These parameters are essential in understanding the architecture and complexity of each model.

In our experiment, each model was standardized with a learning rate of 0.01, a dropout rate of 2, and had four output classes for classification.

The coffee leaves dataset was divided into training, testing, and validation samples. For training the Inception V3, VGG16, ResNet50, Xception, and DenseNet models, 80% of the coffee leaf samples were utilized. Each model underwent ten epochs, and it was observed that all models started to converge with high accuracy after four epochs. The recognition accuracy of the InceptionV3 model is illustrated in Figure 5a, reaching a training accuracy of 99.34%. Figure 5b depicts the log loss of the InceptionV3 model.

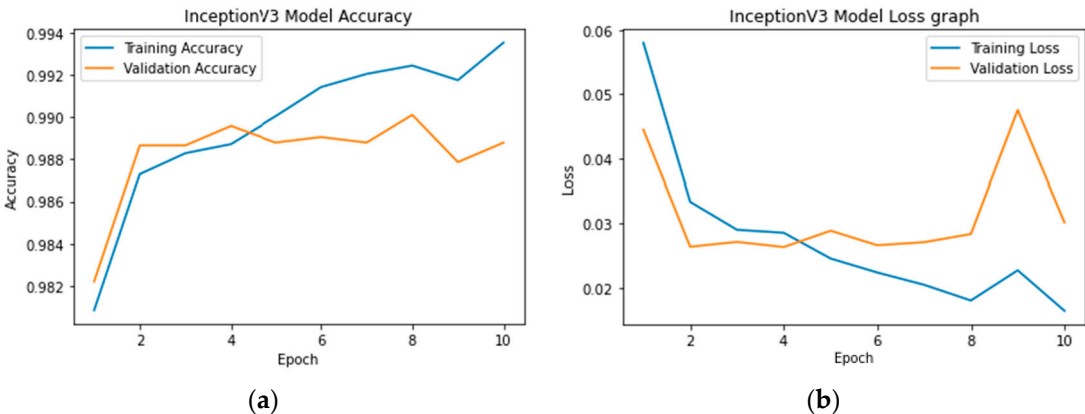

(**a**)　　　　　　　　　　　　　　　　　　　　　　(**b**)

**Figure 5.** InceptionV3 model performance analysis using the collected dataset. (**a**) Model training and validation accuracy; (**b**) Model training and validation loss.

During this research experiment, the second model considered is the ResNet50 model from the same dataset. Following the standardization of hyperparameters, the model

underwent training using 80% of the dataset. Subsequently, 10% of the samples were allocated for testing while the remaining 10% were utilized for validation and testing purposes. From Figure 6a, it can be observed that the model recognition accuracy is around 96% in the first three epochs, and therefore, its stability increased to get an accuracy of 98.70%. This performance is lower than the one represented by InceptionV3 shown in Figure 5. On the other hand, the training and validation losses of the ResNet50 model were around 0.056% and 0.057%, respectively.

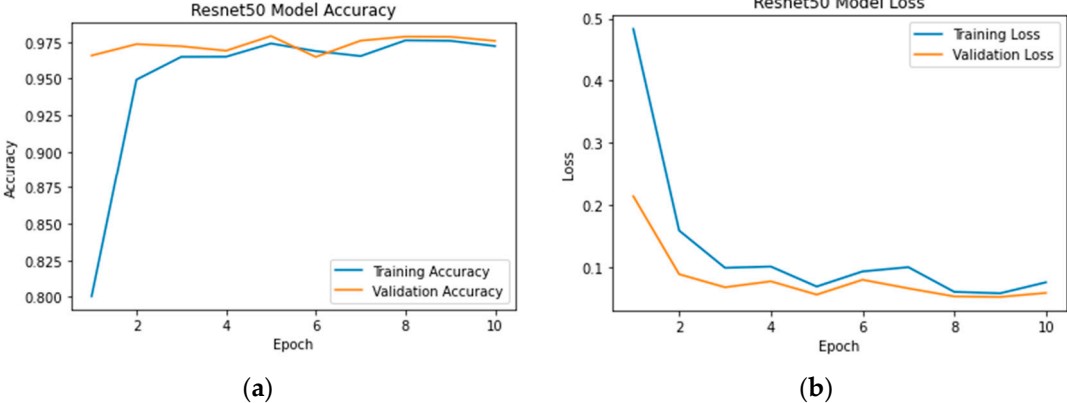

(**a**)                                                       (**b**)

**Figure 6.** ResNet50 model performance analysis using the collected dataset. (**a**) Model training and validation accuracy; (**b**) Model training and validation loss.

Figure 7 demonstrates the behavior of the Xception model on the used datasets after adjusting the hyperparameters. The training and validation accuracy reached 99.40% and 98.84%, respectively, with around four epochs showing less steadiness. Its training and validation losses are shown to be 0.014% and 0.033%, respectively. This execution surpasses that of what the ResNet50 demonstrated, as delineated in Figure 6.

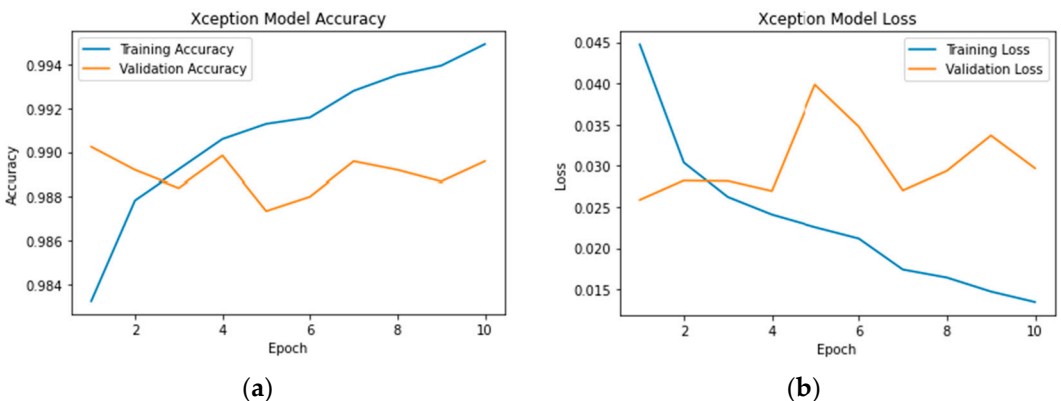

(**a**)                                                       (**b**)

**Figure 7.** Xception model performance analysis using the collected dataset. (**a**) Model training and validation accuracy; (**b**) Model training and validation loss.

The VGG16 model was used as the fourth model using the same dataset. After standardizing the hyperparameters, the model was trained with 80% of the dataset. Subsequently, 10% of the samples were allocated for testing while the remaining 10% were used for validation and testing purposes. By considering Figure 8a, it can be observed that the model achieved a recognition accuracy of approximately 98% in the initial four epochs, and it gradually increased to reach an accuracy of 98.81%. This performance is less than that of the Xception model, as depicted in Figure 6. Furthermore, the training and validation losses of the VGG16 model were approximately 0.0291% and 0.066%, respectively.

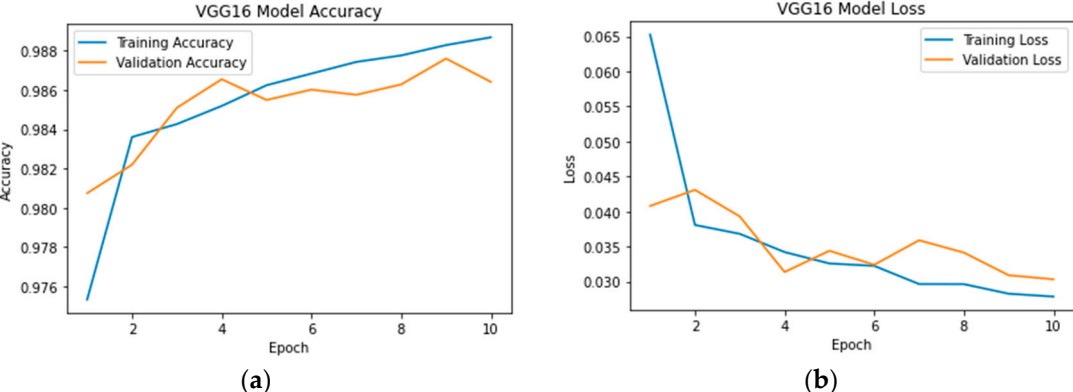

**Figure 8.** VGG16 model performance analysis using the collected dataset. (**a**) Model training and validation accuracy; (**b**) Model training and validation loss.

Figure 9 demonstrates the behavior of the DenseNet model on the used datasets after adjusting the hyperparameters. The training and validation accuracy reached 99.57% and 99.09%, respectively, with around four epochs showing less steadiness. Its training and validation losses are shown to be 0.0135% and 0.0225%, respectively. This execution surpasses that of all other demonstrated models.

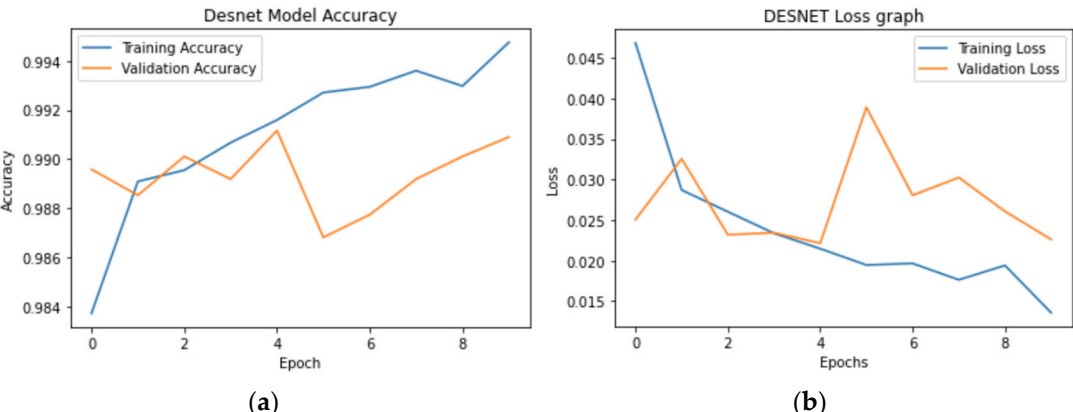

**Figure 9.** DenseNet model performance analysis using the collected dataset. (**a**) Model training and validation accuracy; (**b**) Model training and validation loss.

Figure 10 depicts the behaviors of all five used models on the collected dataset of coffee leaf diseases using Receiver Operating Characteristic (ROC) Curves. It is used to understand indicators of categorization sorting and computational modeling challenges. The curves feature true positive rate (TPR) on the Y axis and false positive rate (FPR) on the X-axis.

It illustrates how the true positive rate (the percentage of correctly classified lesion images) and false positive rate (the percentage of incorrectly classified non-lesion images) change as the classifier's threshold for distinguishing between lesions and non-lesions is adjusted while evaluating test set images.

Figure 11 illustrates the performance of the five employed models on the gathered coffee leaf diseases dataset using precision–recall curves. These curves help serve as a measure to assess the effectiveness of a classifier, especially in situations where there is a significant class imbalance. These curves depict the balance between precision, which gauges the relevance of results, and recall, which measures the comprehensiveness of the classifier's performance.

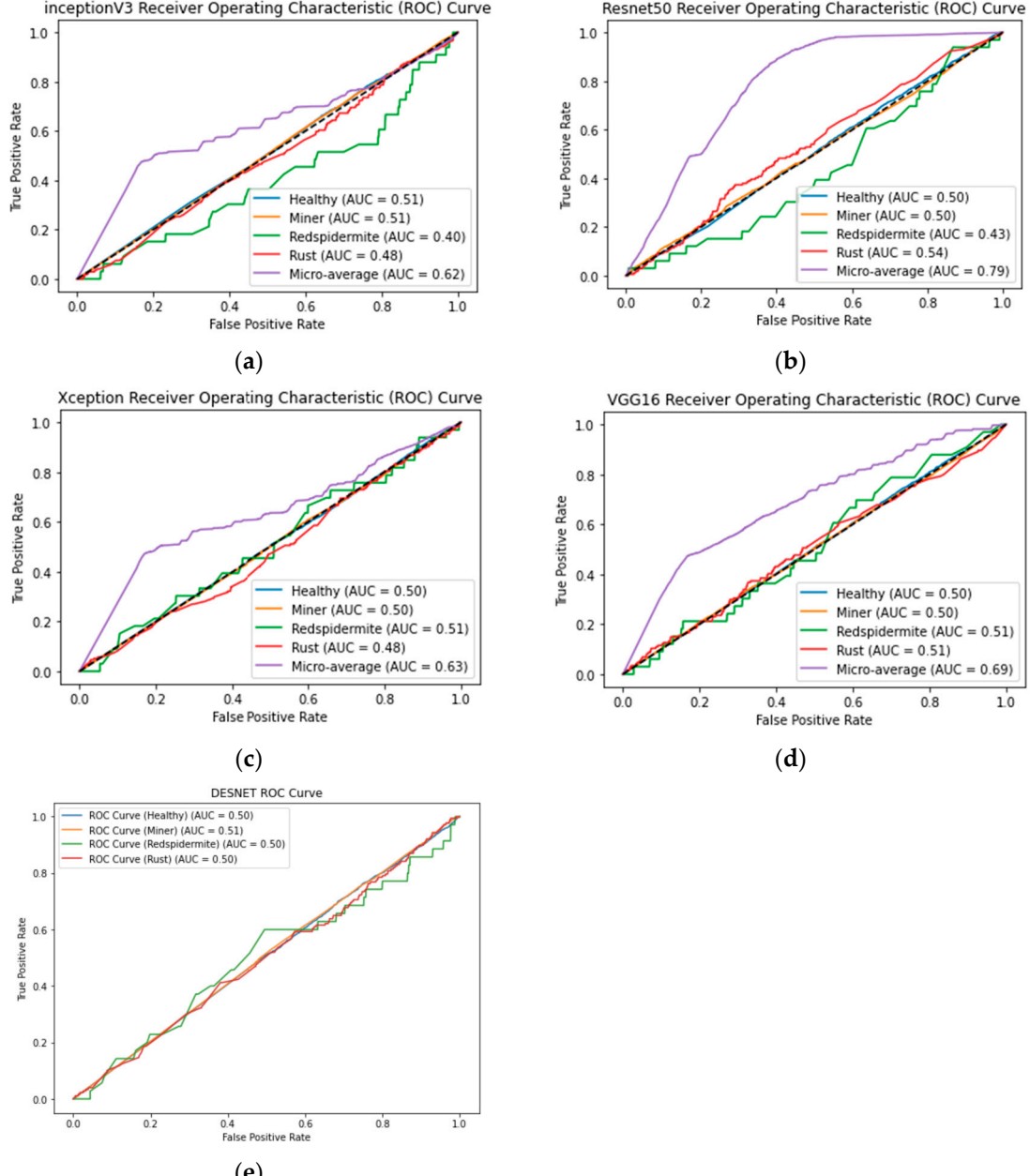

**Figure 10.** Receiver Operating Characteristic (ROC) Curves. (**a**) Details the behaviors of the InceptionV3 model; (**b**) Details the behaviors of the ResNet model; (**c**) Details the behaviors of the Xception model; (**d**) Details the behaviors of the VGG16 model; (**e**) Demonstrates the behaviors of the DenseNet model.

Figure 12 depicts the performance comparison of the five employed models on the gathered coffee leaf diseases dataset using F1 score and MCC metrics. The graph shows the efficiency of the DenseNet Model with an F1 score and MCC of 0.98 and 0.94, respectively. The second proven model is to be VGG16 with an F1 score and MCC of 0.9 and 0.89, respectively. The worst model on the used dataset is shown to be Xception with the F1 score and MCC of 0.48 and 0.4, respectively.

Table 3 provides a comparison of different network models based on their training and validation performance.

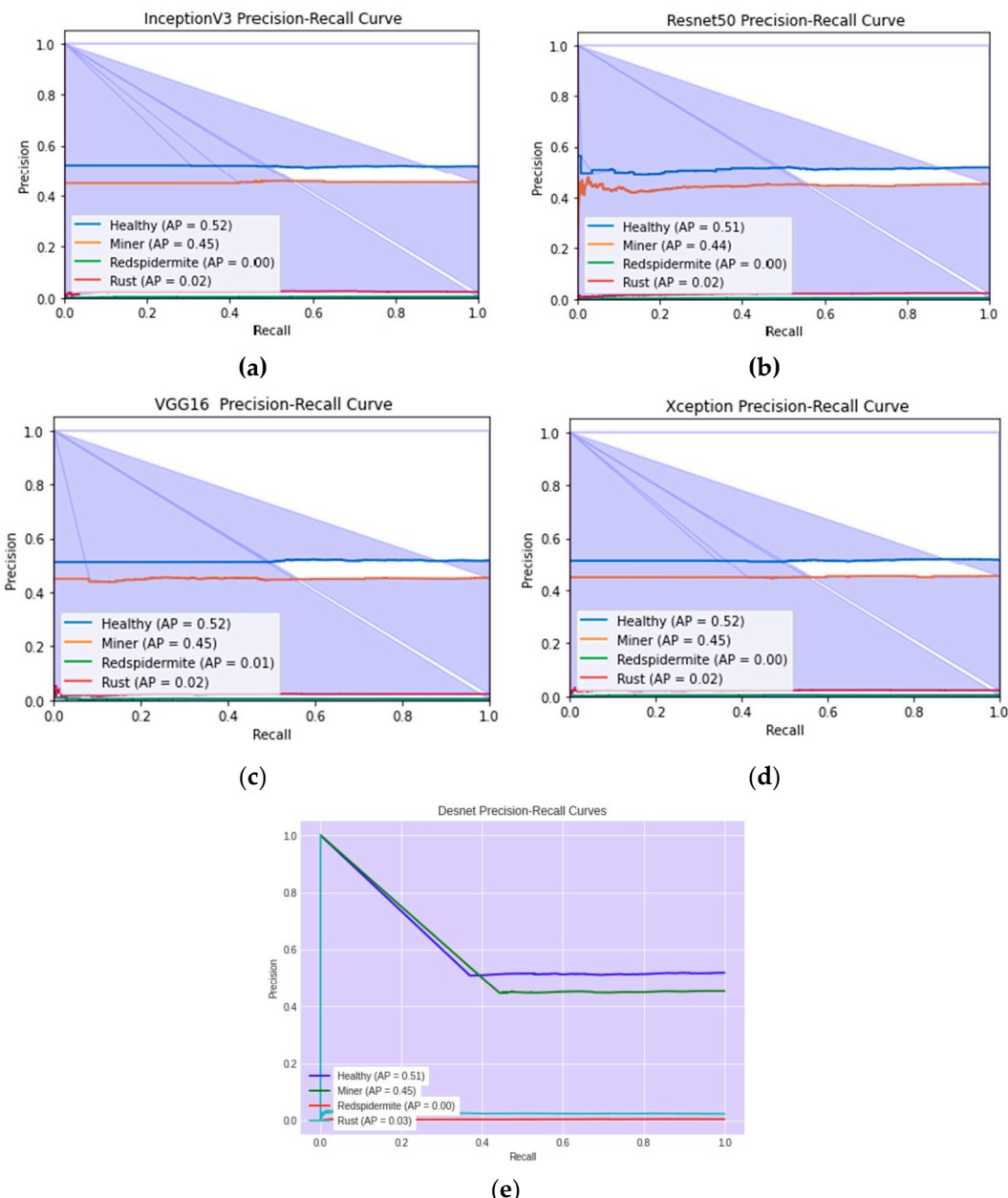

**Figure 11.** Precision–Recall curves of the tested models. (**a**) Details the behaviors of the InceptionV3 model; (**b**) Details the behaviors of the ResNet model; (**c**) Details the behaviors of the Xception model; (**d**) Details the behaviors of the VGG16 model; (**e**) Demonstrates the behaviors of the DenseNet model.

**Table 3.** Summary of network models comparison of performance analysis from the coffee leaf dataset.

| Network Models | Training Accuracy (%) | Training Loss (%) | Validation Accuracy (%) | Validation Loss (%) |
|---|---|---|---|---|
| InceptionV3 | 99.34 | 0.0167 | 99.01 | 0.0306 |
| ResNet50 | 98.70 | 0.0565 | 97.80 | 0.0577 |
| Xception | 99.40 | 0.0140 | 98.84 | 0.0337 |
| VGG16 | 98.81 | 0.0291 | 97.53 | 0.0668 |
| DenseNet | 99.57 | 0.0135 | 99.09 | 0.0225 |

Regarding statistical examination, the ANOVA (Analysis of Variance) test has been executed, and the outcomes are exhibited in Table 4.

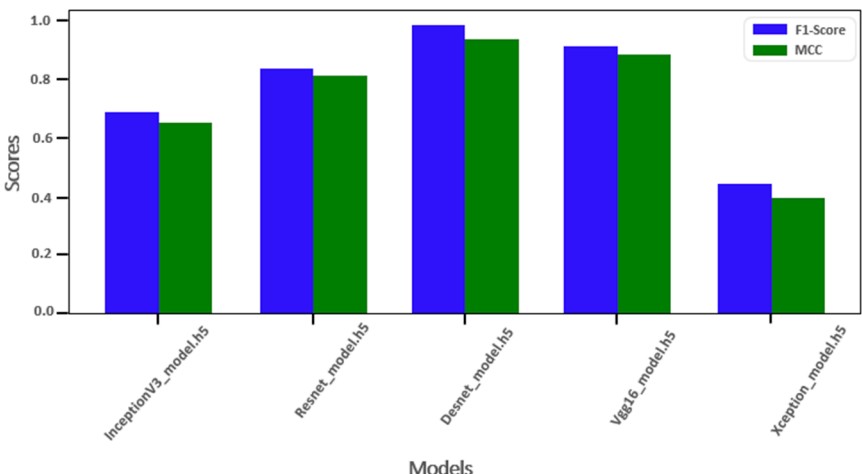

**Figure 12.** Models' comparison using F1-Score and MCC.

**Table 4.** The results of the analysis of variance test.

| ANOVA Table | SS | DF | MS | F-Value | *p*-Value |
|---|---|---|---|---|---|
| Treatment (between columns) | 0.029 | 4 | 0.007 | 233.3333 | $p < 0.0001$ |
| Residual (within columns) | 0.002 | 75 | 0.0003 | | |
| Total | 0.031 | 79 | | | |

The outcomes shown in Table 4 reveal a noteworthy distinction in the selected deep learning algorithm compared to the other methods. This is evident from the ANOVA results provided. The "Treatment" row (which corresponds to the differences between columns) exemplifies this with a substantial F-value of 233.333 and an extremely low *p*-value, less than 0.0001. It is noteworthy that the residual variance is merely 0.002, indicating limited variability among the diverse methods. This suggests that the variation observed in the outcome measure is primarily attributed to the effect of the chosen technique. The variance in the outcome measure was computed across all groups and amounted to 0.031 in the sum of squares. While the ANOVA outcomes point to the superior performance of the selection algorithm concerning the outcome measure compared to other methods, it is important to acknowledge that this is a preliminary observation.

The findings do not provide insights into the magnitude or direction of the effect, nor do they elucidate the specific differences between DenseNet and alternative methods. To ascertain if two samples are extracted from a common population, one can employ a non-parametric method known as the Wilcoxon signed-rank test.

The outcomes of this examination are exhibited in Table 5. Within this table, the assessment aimed to compare the efficacy of the presented models on the dataset.

**Table 5.** The results of the Wilcoxon signed-rank test.

| | DTO + DT | PSO + DT | GWO + DT | GA + DT |
|---|---|---|---|---|
| Theoretical median | $5.75 \times 10^{-8}$ | $5.75 \times 10^{-8}$ | $5.75 \times 10^{-8}$ | $5.75 \times 10^{-8}$ |
| Actual median | $3.57 \times 10^{-5}$ | $3.57 \times 10^{-5}$ | $3.57 \times 10^{-5}$ | $3.57 \times 10^{-5}$ |
| Number of values | 37,964 | 37,964 | 37,964 | 37,964 |
| Wilcoxon Signed-Rank Test | 0 | 0 | 0 | 0 |
| Sum of signed ranks (W) | 37,891 | 37,891 | 37,891 | 37,891 |
| Sum of positive ranks | 1,682,355 | 1,682,355 | 1,682,355 | 1,682,355 |
| Sum of negative ranks | $-1,644,464$ | $-1,644,464$ | $-1,644,464$ | $-1,644,464$ |
| *p*-value (two-tailed) | 0 | 0 | 0 | 0 |
| Exact or estimate? | Exact | Exact | Exact | Exact |
| Significant (alpha = 0.05)? | Yes | Yes | Yes | Yes |
| How big is the discrepancy? | $3.56 \times 10^{-5}$ | $5.06 \times 10^{-8}$ | $9.14 \times 10^{-8}$ | $4.83 \times 10^{-8}$ |



In our comprehensive assessment of the five deep learning models for image classification, we conducted an in-depth analysis to discern their unique capabilities on top of different optimization methods. The results, presented in Table 5, reveal subtle distinctions among these models. Notably, statistical tests, including the Wilcoxon Signed-Rank Test, indicate statistically significant differences in their median performance scores. However, it is crucial to emphasize that these differences, while statistically significant, are practically negligible. Each of the five models, namely InceptionV3, ResNet, DenseNet, VGG16, and Xception, consistently delivered competitive results, reflecting the maturity and robustness of contemporary deep learning architectures. Our study highlights nuanced performance differences while emphasizing the pivotal balance between statistical significance and practical utility, ultimately leading us to select DenseNet as the optimal choice for our image classification task. Nevertheless, it is essential to acknowledge the overall excellence demonstrated by each model, showcasing the prowess of contemporary deep-learning techniques.

## 5. Discussion

In the farming industry, especially for coffee plantations, caring about the importance of coffee consumption worldwide and the drawbacks of coffee diseases and pests affecting production, timely detection of diseases is crucial for achieving high yields. To support improving productivity, the incorporation of the latest technologies is needed for the early diagnosis of coffee diseases from leaves. The literature survey suggested that using deep learning models contributed efficiently to image classification while transfer learning-based models are effective in reducing training computation complexity by addressing the need for extensive datasets. Therefore, this study reveals the application of five pre-trained models in the Rwandan coffee leaf disease dataset to measure performances and provide advice for portable hand-held devices to facilitate farmers.

The performances of models, such as Inception V3, Xception, VGG-16, ResNet-50, and DenseNet, are evaluated with different metrics to identify the most suitable model for the accurate classification of coffee plant leaf diseases. The evaluation metrics, such as ROC, and precision–recall values, were measured.

Figure 9 illustrates a graphical representation of the pre-trained network models based on the evaluation metric, such as ROC. The VGG16 and DenseNet present good performance compared to other used models on all disease classes. The AUC for all discussed diseases in this context appeals to be in the range of 0.5 to 1. This indication means that the model can correctly classify coffee rust, minor, health, and red spider mites surveyed to be abundant in Rwanda. To tackle the problem of vanishing gradients induced by skip connections, we utilized regularization methods, such as batch normalization. The use of deeper models presented several difficulties, such as overfitting, covariant shifts, and longer training times. To surmount these obstacles, we conducted experiments to finely adjust the hyperparameters.

The assessment of model performance was measured using the AP metric as shown in Equation (3). In the performed experiment of the dataset used on the selected pre-trained models, Figure 10 shows the results of different models. The illustration demonstrated that DenseNet and VGG16 have better AP for the used classes than InceptionV3, Xception, and ResNet50. DenseNet demonstrates AP values of 51%, 40%, 0%, and 3% for health, miner, and red spider mite class, respectively. VGG16 demonstrates AP values of 52%, 45%, 1%, and 2% for health, miner, and red spider mite class, respectively. The VGG16 expressed to grab some detections on red spider mites compared to others. The observation is that lack of enough images in this class. The evaluation outcomes revealed that DenseNet and VGG16 performed better than InceptionV3, Xception, and ResNet50 models.

Table 1 presents different research references, the year of publication, the methods used, accuracy percentages, and the corresponding plant names for leaf classification. The "proposed model" labeled as DenseNet achieved the highest accuracy of 99.57% in classifying coffee leaves. Table 3 shows the comparison of different models and their score

accuracies. The training accuracy and loss represent how well the models performed on the training data while the validation accuracy and loss show their performance on previously unseen validation data. Among the models, DenseNet achieved the highest training accuracy (99.57%) and validation accuracy (99.09%), indicating its excellent ability to learn and generalize from the data. On the other hand, ResNet50 had the lowest validation accuracy (97.80%) and the highest validation loss (0.0577), suggesting it might slightly struggle to generalize to new data compared to the other models. To emphasize the model evaluation criteria, we performed statistical tests with ANOVA and Wilcoxon, as shown in Tables 4 and 5, to check the variability of models on our dataset. It reaffirms our decision to choose the 'DenseNet' model based on a comprehensive evaluation of various factors, including not only ANOVA or Wilcoxon tests, but also median discrepancies and other metrics discussed.

## 6. Conclusions and Future Directions

In this study, we investigated the coffee farming industry in Rwanda, focusing on various identified coffee leaf diseases. Our research involved a successful analysis of different transfer learning models, specifically chosen to accurately classify five distinct classes of coffee plant leaf diseases. We standardized and evaluated cutting-edge deep learning models using transfer learning techniques, considering the classification accuracy, precision, recall, and AP score as the evaluation metrics. After analyzing several pre-trained architectures, including InceptionV3, Xception, and ResNet50, we found that DenseNet and VGG16 performed exceptionally well. Based on our findings, we proposed a model training pipeline that was followed throughout the experiment.

DenseNet model training was found to be more straightforward, primarily attributed to its smaller number of trainable parameters and lower computational complexity. This quality makes DenseNet particularly well-suited for coffee plant leaf disease identification, especially when incorporating new coffee leaf diseases that were not part of the initial training data, as it reduces the overall training complexity. The experimented model's quality has been tested using statistical tests, such as Wilcoxon and ANOVA. The proposed model demonstrated exceptional performance, achieving an impressive classification accuracy of 99.57%, along with high values for AUC and AP metrics.

In our future endeavors, we aim to tackle challenges associated with real-time data collection. We plan to develop a multi-object deep learning model capable of detecting coffee plant leaf diseases not just from individual leaves, but also from a bunch of leaves as well. Moreover, we are currently working on the implementation of a mobile application that will leverage the trained model obtained from this study. This application will provide valuable assistance to farmers and the agricultural sector by enabling the real-time identification of leaf diseases in Rwanda based on the samples taken.

**Author Contributions:** Conceptualization, E.H. and G.B.; methodology, E.H., E.M., G.B., S.M.M. and P.R.; software, E.H., G.B. and J.N.; validation, E.H., J.N. and G.B.; formal analysis, E.H., S.M.M. and O.J.S.; investigation, E.H., M.C.A.K., J.M., E.M., J.A.U.U., L.C.C. and T.M.; resources, E.H.; data curation, J.N.; writing—original draft preparation, E.H.; writing—review and editing, E.H., G.B., J.C.U. and M.C.A.K.; visualization, J.N.; supervision, G.B. and P.R.; project administration, E.H.; funding acquisition, O.J.S. and P.R. All authors have read and agreed to the published version of the manuscript.

**Funding:** This research was funded through a grant offered by the University of Rwanda in partnership with SIDA (Swedish International Development Agency) under the programme UR-Sweden program. The grant supported all research activities, such as data collection, purchase of equipment and materials, fieldwork, etc. The APC was also funded by the same grant.

**Institutional Review Board Statement:** Not applicable.

**Informed Consent Statement:** Not applicable.

**Data Availability Statement:** When requested, the authors will make available all data used in this study.

**Acknowledgments:** This work is acknowledged by the Rwanda Agricultural Board (RAB) to avail the farming cooperatives operating in Rwanda and the coffee washing stations.

**Conflicts of Interest:** The authors declare no conflict of interest.

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
