# Peer review of "An Intelligent System-Based Coffee Plant Leaf Disease Recognition Using Deep Learning Techniques on Rwandan Arabica Dataset"

_technologies, doi:10.3390/technologies11050116_

Round 1
Reviewer 1 Report
few minor changes required.
Very good topic selected.

only on instance of grammar detected.
Author Response
Responses are attached.
Regards

Reviewer 2 Report
This work aims to build the Rwandan coffee plant dataset, with coffee rust, miner, and red spider mite identified as the most popular due to their geographical situations. From the collected coffee leaves dataset of 37,939 images, the preprocessing and modeling used five deep learning models InceptionV3, ResNet50, Xception, VGG16, and DenseNet. The paper's contribution to existing knowledge in this research field is justified. The article needs to contribute more; the following points can improve the manuscript.
1. The introduction should be shorter; divide it into the Introduction and Related Work sections.
2. A comparative study can be added to a related work section in table form to show the recent efforts.
3. It needs to show your innovations and contributions clearly. Please highlight your innovations.
4. Double-check all the equations to be true.
5. There needs to be more than performance evaluation metrics. Add some other metrics and explain them mathematically.
6. The proposed method should be compared with more recent techniques.
7. Some statistical tests, such as Wilcoxon and ANOVA, should be performed to ensure the quality of the presented methods.
8. Improve the English of the work. Proofreading is recommended.
9. Change the “Conclusion” section title to “Conclusion and Future Directions."
Proofreading is recommended.
Author Response
Responses are attached.
Regards
